

# Semi-supervised oblique predictive clustering trees

Tomaž Stepišnik[1,2] and Dragi Kocev[1,2,3]

[1] Department of Knowledge Technologies, Jožef Stefan Institute, Ljubljana, Slovenia
[2] Jožef Stefan International Postgraduate School, Ljubljana, Slovenia
[3] Bias Variance Labs, Ljubljana, Slovenia

## ABSTRACT

Semi-supervised learning combines supervised and unsupervised learning approaches to learn predictive models from both labeled and unlabeled data. It is most appropriate for problems where labeled examples are difficult to obtain but unlabeled examples are readily available (e.g., drug repurposing). Semi-supervised predictive clustering trees (SSL-PCTs) are a prominent method for semi-supervised learning that achieves good performance on various predictive modeling tasks, including structured output prediction tasks. The main issue, however, is that the learning time scales quadratically with the number of features. In contrast to axis-parallel trees, which only use individual features to split the data, oblique predictive clustering trees (SPYCTs) use linear combinations of features. This makes the splits more flexible and expressive and often leads to better predictive performance. With a carefully designed criterion function, we can use efficient optimization techniques to learn oblique splits. In this paper, we propose semi-supervised oblique predictive clustering trees (SSL-SPYCTs). We adjust the split learning to take unlabeled examples into account while remaining efficient. The main advantage over SSL-PCTs is that the proposed method scales linearly with the number of features. The experimental evaluation confirms the theoretical computational advantage and shows that SSL-SPYCTs often outperform SSL-PCTs and supervised PCTs both in single-tree setting and ensemble settings. We also show that SSL-SPYCTs are better at producing meaningful feature importance scores than supervised SPYCTs when the amount of labeled data is limited.

## INTRODUCTION

The most common tasks in machine learning are supervised and unsupervised learning. In supervised learning, we are presented with a set of examples described with their properties (i.e., descriptive variables or features) as well as with a target property (i.e., output variables, target variables, or labels). The goal of a supervised learning method is to learn a mapping from the descriptive values to the output values that generalizes well to examples that were not used for learning. In unsupervised learning, on the other hand, no output values are provided for the examples. Instead, unsupervised methods aim to extract some underlying

Corresponding author
Tomaž Stepišnik,
tomaz.stepisnik@ijs.si

structure of the examples (e.g., discover clusters of similar examples, learn low dimensional representations, etc.).

Semi-supervised learning combines these two approaches (*Chapelle, Schölkopf & Zien, 2006*). We are presented with a set of examples, where a (smaller) part of them are associated with output values (labeled examples), and a (larger) part of them are not (unlabeled examples). Semi-supervised methods learn a mapping from examples to the output values (like supervised methods), but also include unlabeled examples in the learning process (like unsupervised methods). The semi-supervised approach is typically used when learning examples are too scarce for supervised methods to learn a model that generalizes well, and, at the same time, unlabeled examples are relatively easy to obtain. This often happens in problems from life sciences, where the process of labeling the examples requires wet-lab experiments that are time-consuming and expensive. For example, consider the problem of discovering a new drug for a certain disease. Testing the effects of compounds on the progression of the disease requires screening experiments, so labeling the examples (compounds) is expensive. On the other hand, millions of unlabeled compounds are present and described in online databases. Ideally, a semi-supervised method can use a handful of labeled compounds, combine them with the unlabeled compounds, and learn a model that can predict the effect of a compound on the disease progression, to facilitate the discovery of a novel drug.

The most common approaches to semi-supervised learning are wrapper methods (*Van Engelen & Hoos, 2020*), such as self-training (*Kang, Kim & Cho, 2016*), where a model iteratively labels the unlabeled examples and includes these pseudo-labels in the learning set in the next iteration. Alternatively, in co-training (*Zhou & Li, 2007*) there are two models that iteratively label the data for each other. Typically, these two models are different or at least learn on different views of the data. Among the intrinsically semi-supervised methods (*Van Engelen & Hoos, 2020*), semi-supervised predictive clustering trees (*Levatić, 2017*) are a prominent method. They can be used to solve a variety of predictive tasks, including multi-target regression and (hierarchical) multi-label classification (*Levatić, 2017*; *Levatić et al., 2017*; *Levatić et al., 2018*; *Levati et al., 2020*). They achieve good predictive performance and, as a bonus, the learned models can be interpreted, either by inspecting the learned trees or calculating feature importances from ensembles of trees (*Petkovi, Deroski & Kocev, 2020*). However, the method scales poorly with data dimensionality—the model learning can take a very long time on datasets with many features or targets.

Standard decision/regression trees (*Breiman et al., 1984*) split data based on the features in a way that minimizes the impurity of the target in the resulting clusters (e.g., variance for regression, entropy for classification). In the end nodes (leaves), predictions for the target are made. Predictive clustering trees (*Blockeel, Raedt & Ramon, 1998*; *Blockeel et al., 2002*) (PCTs) generalize standard trees by differentiating between three types of attributes: features, clustering attributes, and targets. Features are used to divide the examples; these are the attributes encountered in the split nodes. Clustering attributes are used to calculate the heuristic that guides the search of the best split at a given node, and targets are predicted in the leaves. The role of the targets in standard trees is therefore split between the clustering attributes and targets in PCTs. In theory, the clustering attributes can be

selected independently of the features and the targets. However, the learned tree should make accurate predictions for the targets, so minimizing the impurity of the clustering attributes should help minimize the impurity of the targets. This attribute differentiation gives PCTs a lot of flexibility. They have been used for predicting various structured outputs (*Kocev et al., 2013*), including multi-target regression, multi-label classification, and hierarchical multi-label classification. Embeddings of the targets have been used as clustering attributes in order to reduce the time complexity of tree learning (*Stepišnik & Kocev, 2020a*). Semi-supervised PCTs use both targets and features as clustering attributes. This makes leaves homogeneous in both the input and the output space, which allows unlabeled examples to influence the learning process.

PCTs use individual features to split the data, which means the split hyperplanes in the input spaces are axis-parallel. SPYCTs (*Stepišnik & Kocev, 2020b*; *Stepinik & Kocev, 2020*) are a redesign of standard PCTs and use linear combinations of features to achieve oblique splits of the data—the split hyperplanes are arbitrary. The potential advantage of oblique splits compared to axis-parallel splits is presented in Fig. 1. SPYCTs offer state-of-the-art predictive performance, scale better with the number of clustering attributes, and can exploit sparse data to speed up computation.

In this paper, we propose SPYCTs for semi-supervised learning. We follow the same semi-supervised approach that regular PCTs do, which includes features in the heuristic function for evaluating the quality of a split. This makes the improved scaling of SPYCTs over PCTs especially beneficial, which is the main motivation for our proposal. We modify the oblique split learning objective functions of SPYCTs to account for missing target values. We evaluate the proposed approach on multiple benchmark datasets for different predictive modeling tasks.

In the remainder of the paper, we first describe the proposed semi-supervised methods and present the experimental setting for their evaluation. Next, we present and discuss the results of our experiments and, finally, conclude the paper by providing several take-home messages.

## METHOD DESCRIPTION

In this section, we present our proposal for semi-supervised learning of SPYCTs (SSL-SPYCTs). We start by introducing the notation used in the manuscript. Let $X^l \in \mathbb{R}^{L \times D}$ and $X^u \in \mathbb{R}^{U \times D}$ be the matrices containing the $D$ features of the $L$ labeled and $U$ unlabeled examples, respectively. Let $Y \in \mathbb{R}^{L \times T}$ be the matrix containing the $T$ targets associated with the $L$ labeled examples. And let $X = [(X^l)^T (X^u)^T]^T \in \mathbb{R}^{(L+U) \times D}$ be the matrix combining the features of both labeled and unlabeled examples. Finally, let $p \in \mathbb{R}^{D+T}$ be the vector of clustering weights, used to put different priorities to different clustering attributes (features and targets) when learning a split.

There are two variants of SPYCTs that learn the split hyperplanes in different ways.

1. The *SVM variant* first groups the examples into two clusters based on the clustering attributes using $k$-means clustering, then learns a linear SVM on the features with cluster indicators as targets to approximate this split.

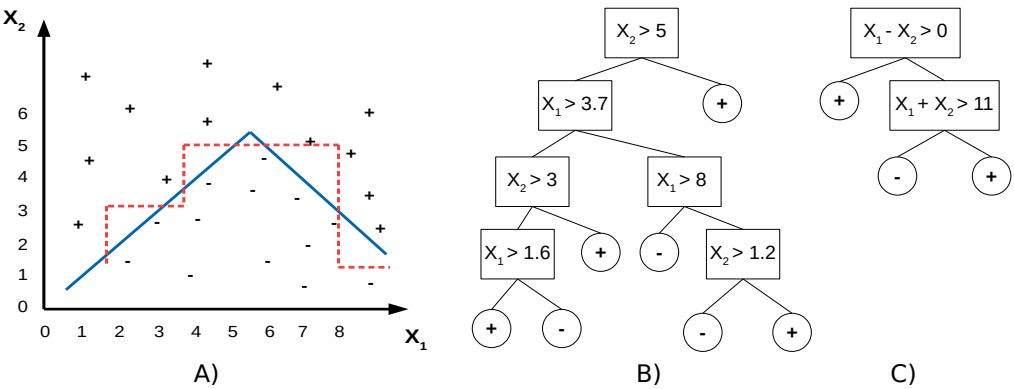

**Figure 1** A toy dataset (A) with drawn decision boundaries learned by the axis-parallel (red, dashed) and oblique (blue, solid) decision trees (B, C).

2. The *gradient variant* uses a fuzzy membership indicator to define a differentiable objective function which measures the impurity on both sides of the split hyperplane. The hyperplane is then optimized with gradient descent to minimize the impurity.

The basis of the semi-supervised approach is to use both features and targets as clustering attributes, so that unlabeled examples influence the learning process through the heuristic score calculation, despite the missing target values. For the SVM variant, this means that examples are clustered based on both target and feature values. For the gradient variant, the split is optimized to minimize the impurity of both features and targets on each side of the hyperplane. The overview of the SSL-SPYCT learning algorithm is presented in Algorithm 1. The weights $w \in \mathbb{R}^D$ and bias $b \in \mathbb{R}$ define the split hyperplane, and they are obtained differently for each SPYCT variant as follows.

---

**Algorithm 1** Learning a SSL-SPYCT: The inputs are features $X_l \in \mathbb{R}^{L \times D}$ and $X_u \in \mathbb{R}^{U \times D}$ of labeled and unlabeled examples, targets $Y \in \mathbb{R}^{L \times T}$ of the labeled examples, and a vector $c \in \mathbb{R}^{D+T}$ of clustering weights.

1: **procedure** GROW_TREE($X_l$, $X_u$, $Y$, $c$)
2:     $w, b = \text{get\_split\_hyperplane}(X_l, X_u, Y, c)$
3:     score $= Xw + b$
4:     rows1 $= \{i \,|\, \text{score}_i > 0\}$
5:     rows2 $= \{i \,|\, \text{score}_i \leq 0\}$
6:     **if** acceptable_split(rows1, rows2) **then**
7:         left_subtree $= \text{grow\_tree}(X_l[\text{rows1}], X_u[\text{rows1}], Y[\text{rows1}], c)$
8:         right_subtree $= \text{grow\_tree}(X_l[\text{rows2}], X_u[\text{rows2}], Y[\text{rows2}], c)$
9:         **return** Node($w$, $b$, left_subtree, right_subtree)
10:    **else**
11:        **return** Leaf(prototype($Y$))

---

**SVM variant** The first step is to cluster examples into two groups using $k$-means clustering. The initial centroids are selected randomly from the labeled examples. Since the clustering is performed based on both features and targets, the cluster centroids consist of feature and target parts, i.e.,

$$c^0 = \begin{bmatrix} X_{i,:}^l & Y_{i,:} \end{bmatrix} \in \mathbb{R}^{D+T}, \quad c^1 = \begin{bmatrix} X_{j,:}^l & Y_{j,:} \end{bmatrix} \in \mathbb{R}^{D+T}.$$

Next, we calculate the Euclidean distance to the two centroids for each of the examples. For unlabeled examples, we only calculate the distance to the feature part of the centroids ($c_0^x$ and $c_1^x$):

$$d(i,j) = \sum_{k=1}^{D} p_k (X_{j,k} - c_k^i)^2 + \alpha \sum_{k=1}^{T} p_{D+k} (Y_{j,k} - c_{D+k}^i)^2,$$

where $i \in \{0,1\}$ is the cluster indicator, $1 \leq j \leq L+U$ is the example index, and $\alpha = 1$ if the example is labeled (i.e., $j \leq L$) and $\alpha = 0$ if it is unlabeled. The examples are split into two clusters according to the closer centroid. In the case of ties in the distance, the examples are assigned (uniformly) randomly to a cluster.

Let $s \in \{0,1\}^{L+U}$ be the vector indicating the cluster membership. The new centroids are then the means of the examples assigned to each cluster. The means of the target parts of the centroids are calculated only using the labeled examples, i.e.,

$$c_j^i = \frac{\sum_{k=1}^{L+U} \mathbb{I}[s_k = i] X_{k,j}}{\sum_{k=1}^{L+U} \mathbb{I}[s_k = i]}, \quad \text{if } 1 \leq j \leq D,$$

$$c_j^i = \frac{\sum_{k=1}^{L} \mathbb{I}[s_k = i] Y_{k,j-D}}{\sum_{k=1}^{L} \mathbb{I}[s_k = i]}, \quad \text{if } D < j \leq D+T.$$

This procedure is repeated for a specified number of iterations. After the final clusters are determined, a linear SVM is used to approximate this split based on the features. Specifically, the following optimization problem is solved:

$$\min_{w,b} ||w||_1 + C \sum_{k=1}^{L+U} max(0, 1 - s_k(X_{k,:} \cdot w + b))^2,$$

where parameter $C \in \mathbb{R}$ determines the strength of regularization.

**Gradient variant** We start with randomly initialized weights ($w$) and bias ($b$) and calculate the fuzzy membership vector $s = \sigma(Xw + b) \in [0,1]^{L+U}$. The value $s_i$ tells us how much the corresponding example belongs to the "positive" group, whereas the value $1 - s_i$ tells us how much it belongs to the "negative" group.

To calculate the impurity of a group, we calculate the weighted variance for every feature and every target. For the targets, only labeled examples are used in the calculation. Weighted variance of a vector $v \in \mathbb{R}^n$ with weights $a \in \mathbb{R}^n$ is defined as

$$var(v,a) = \frac{\sum_i^n a_i(v_i - mean(v,a))^2}{A} = mean(v^2,a) - mean(v,a)^2,$$

where $A = \sum_i^n a_i$ is the sum of weights and $mean(v,a) = \frac{1}{A}\sum_i^n a_i v_i$ is the weighted mean of $v$. The impurity of the positive group is then calculated as

$$imp(s,p) = \sum_{k=1}^{D} p_k var(X_{:,k},s) + \sum_{k=1}^{T} p_{D+k} var(Y_{:,k},s).$$

To get the impurity of the negative group $imp(1-s,p)$, we simply swap the fuzzy membership weights with $1-s$. The split fitness function we wish to optimize is then

$$f(w,b) = S \cdot imp(s,p) + (L+U-S) \cdot imp(1-s,p),$$

where $s = \sigma(Xw+b)$ and $S = \sum_i s_i$. The terms $S$ and $L+U-S$ represent the sizes of the positive and negative subsets and are added to guide the split search towards balanced splits. The final optimization problem for learning the split hyperplane is

$$\min_{w,b} ||w||_{\frac{1}{2}} + Cf(w,b),$$

where $C$ again controls the strength of regularization. The objective function is differentiable, and we can efficiently solve the problem using the Adam (*Kingma & Ba, 2014*) gradient descent optimization method.

The clustering weights are uniform for the targets for tasks of binary classification, multi-class classification, multi-label classification, regression, and multi-target regression. For hierarchical multi-label classification, the weights for target labels positioned lower in the hierarchy are smaller. This gives more importance to labels higher in the hierarchy when splitting the examples.

Features and clustering attributes are standardized to mean 0 and standard deviation 1 prior to learning each split. For the features, this is done to make split learning more stable. For the clustering attributes, this is performed before the application of the clustering weights, so that only clustering weights control the relative influences of the different clustering attributes on the objective function.

We also implement a parameter $\omega$ that determines the degree of supervision. The clustering weights, corresponding to features ($p_i$ for $1 \leq i \leq D$), are scaled so that their sum is $1-\omega$, and clustering weights, corresponding to targets ($p_i$ for $D < i \leq D+T$, are scaled

so that their sum is $\omega$. This enables us to determine the relative importance of features and targets when splitting the data. With the borderline values selected for $\omega$ (0 or 1), we get the extreme behavior in terms of the amount of supervision. Setting the value of $\omega$ to 0 means that the target impurity is ignored and tree construction is effectively unsupervised, i.e., without supervision. Alternatively, setting the value of $\omega$ to 1 means that feature impurity is ignored when learning splits, hence, the unlabeled examples do not affect the split selection. The tree construction in this case is fully supervised.

The splitting of the examples (i.e., the tree construction) stops when at least one of the following stopping criteria is reached. We can specify the minimum number of examples required in leaf nodes (at least one labeled example is always required otherwise predictions cannot be made). We can also require a split to reduce the impurity by a specified amount or specify the maximum depth of the tree.

After the splitting stops, a leaf node is created. The prototype of the targets of the remaining examples is calculated and it is stored for use as the prediction for the examples reaching that leaf. Since the targets in SOP are represented as tuples/vectors, the prototypes are calculated as column-wise mean values of the targets ($Y$). They can be used directly as predictions (in regression problems) or used to calculate the majority class (in binary and multi-class classification), or used to predict all labels with the mean above a certain threshold (in hierarchical and flat multi-label classification).

The time complexity of learning a split in standard PCTs is $O(DN \log N + NDK)$ (*Kocev et al., 2013*), where $K$ is the number of clustering attributes. For the SVM and gradient variant of SPYCTs, the time complexities are $O(N(I_c K + I_o D))$ and $O(NI_o(D + K))$, respectively (*Stepinik & Kocev, 2020*), where $I_o$ is the number of $w, b$ optimization iterations and $I_c$ is the number of clustering iterations (SVM variant). When learning SSL variants (SSL-PCTs and SSL-SPYCTs), clustering attributes consist of both features and targets, therefore $K = D + T$. This means that SSL-PCTs scale quadratically with the number of features, whereas both variants of SSL-SPYCTs scale linearly. SSL-SPYCTs are therefore much more computationally efficient, and can additionally take advantage of sparse data by performing calculations with sparse matrices. Our implementation of the proposed method is freely licensed and available for use and download at https://gitlab.com/TStepi/spyct.

## EXPERIMENTAL DESIGN

We evaluated our approach on 30 benchmark dataset for different predictive modeling tasks: binary classification (BC), multiclass classification (MCC), multi-label classification (MLC), and hierarchical multi-label classification (HMLC), single-target regression (STR) and multi-target regression (MTR). The datasets are freely available and were obtained from the following repositories: `openml` (https://www.openml.org), `mulan` (http://mulan.sourceforge.net/datasets.html), `dtai-cs` (https://dtai.cs.kuleuven.be/clus/hmc-ens/) and `kt-ijs` (http://kt.ijs.si/DragiKocev/PhD/resources/doku.php?id=hmc_classification). The selected datasets have diverse properties in terms of application domains, number of examples, number of features, and number of targets. Their properties and sources are presented in Table 1.

**Table 1** **Details of the benchmark datasets used for the evaluation.** The task column shows the predictive modeling task applicable to the datasets (BC is binary classification, MCC is multi-class classification, MLC is multi-label classification, HMLC is hierarchical multi-label classification, STR is single-target regression, MTR is multi-target regression), N is the number of examples, D is the number of features, and T is the number of targets (for MCC, it is the number of classes).

| dataset | source | task | N | D | T |
|---------|--------|------|------|------|-----|
| bioresponse | openml | BC | 3751 | 1776 | 1 |
| mushroom | openml | BC | 8124 | 22 | 1 |
| phoneme | openml | BC | 5404 | 5 | 1 |
| spambase | openml | BC | 4601 | 57 | 1 |
| speeddating | openml | BC | 8378 | 120 | 1 |
| cardiotocography | openml | MCC | 2126 | 35 | 10 |
| gesture | openml | MCC | 9873 | 32 | 5 |
| isolet | openml | MCC | 7797 | 617 | 26 |
| mfeat-pixel | openml | MCC | 2000 | 240 | 10 |
| plants-texture | openml | MCC | 1599 | 64 | 100 |
| bibtex | mulan | MLC | 7395 | 1836 | 159 |
| birds | mulan | MLC | 645 | 260 | 19 |
| bookmarks | mulan | MLC | 87856 | 2150 | 208 |
| delicious | mulan | MLC | 16105 | 500 | 983 |
| scene | mulan | MLC | 2407 | 294 | 6 |
| ara_interpro_GO | dtai-cs | HMLC | 11763 | 2815 | 630 |
| diatoms | kt-ijs | HMLC | 3119 | 371 | 398 |
| enron | kt-ijs | HMLC | 1648 | 1001 | 56 |
| imclef07d | kt-ijs | HMLC | 11006 | 80 | 46 |
| yeast_seq_FUN | dtai-cs | HMLC | 3932 | 478 | 594 |
| cpmp-2015 | openml | STR | 2108 | 23 | 1 |
| pol | openml | STR | 15000 | 48 | 1 |
| qsar-197 | openml | STR | 1243 | 1024 | 1 |
| qsar-12261 | openml | STR | 1842 | 1024 | 1 |
| satellite_image | openml | STR | 6435 | 36 | 1 |
| atp1d | mulan | MTR | 337 | 411 | 6 |
| enb | mulan | MTR | 768 | 8 | 2 |
| oes97 | mulan | MTR | 334 | 263 | 16 |
| rf2 | mulan | MTR | 9125 | 576 | 8 |
| scm1d | mulan | MTR | 9803 | 280 | 16 |

We focus on the comparison of our proposed SSL-SPYCT method with the original supervised method SPYCT and the semi-supervised learning of axis-parallel PCTs: the SSL-PCT (*Levatić, 2017*). These two baselines are the most related supervised and semi-supervised methods of the proposed approach, respectively. For completeness, we also include supervised PCTs in the comparison. Note that SPYCTs and PCTs are the only available methods able to address all of the structured output prediction tasks in a uniform manner. We evaluate the methods in single tree setting and in bagging ensembles (*Breiman, 1996*) of 50 trees.

For SPYCTs we use the same configuration as it was used in *Stepišnik & Kocev (2020c)*. Tree depth is not limited, leaves only need to have 1 (labeled) example, and splits are accepted if they reduce impurity by at least 5% in at least one of the subsets. The maximum number of optimization iterations is set to 100 for both variants, and the SVM variant uses at most 10 clustering iterations. The strength of regularization ($C$) is set to 10. For the gradient variant, the Adam optimizer uses parameters $\beta_1 = 0.9$, $\beta_2 = 0.999$, and $\epsilon = 10^{-8}$. These are the default values from the PyTorch (https://pytorch.org/docs/1.1.0/_modules/torch/optim/adam.html) library.

For semi-supervised methods, we select the $\omega$ parameter with 3-fold internal cross-validation on the training set. We select the best value from the set $\{0, 0.25, 0.5, 0.75, 1\}$. We investigate the influence of the number of labeled examples $L$ on the performance of the semi-supervised methods. We set $L$ to the following numbers of available labeled examples: $\{25, 50, 100, 250, 500\}$. We evaluate the methods with a slightly modified 10-fold cross-validation corresponding to inductive evaluation setting. First, a dataset is divided into 10 folds. One fold is used as the test set. From the other 9 folds, $L$ examples are randomly selected as labeled examples, and the rest are used as unlabeled examples. This process is repeated 10 times so that each fold is used once as the test set. On the two MTR datasets that have fewer than 500 examples (*atp1d* and *oes97*) experiments with $L = 500$ are not performed.

To measure the predictive performance of the methods on STR and MTR datasets, we use the *coefficient of determination*

$$R^2(y, \hat{y}) = 1 - \frac{\sum_i (y_i - \hat{y}_i)^2}{\sum_i (y_i - \bar{y})^2},$$

where $y$ is the vector of true target values, $\bar{y}$ is their mean, and $\hat{y}$ is the vector of predicted values. For MTR problems, we calculate the mean of $R^2$ scores per target. For BIN and MCC tasks, we use *F1 score*, macro averaged in the MCC case.

Methods solving MLC and HMLC tasks typically return a score for each label and each example, a higher score meaning that an example is more likely to have that label. Let $y \in \{0, 1\}^{n \times l}$ be the matrix of label indicators and $\hat{y} \in \mathcal{R}^{n \times l}$ the matrix of label scores returned by a method. We measured the performance of methods with weighted label ranking average precision

$$LRAP(y, \hat{y}) = \frac{1}{n} \sum_{i=0}^{n-1} \sum_{j: y_{ij} = 1} \frac{w_j}{W_i} \frac{L_{ij}}{R_{ij}},$$

where $L_{ij} = |\{k : y_{ik} = 1 \wedge \hat{y}_{ik} \geq \hat{y}_{ij}\}|$ is the number of real labels assigned to example $i$ that the method ranked higher than label $j$, $R_{ij} = |\{k : \hat{y}_{ik} \geq \hat{y}_{ij}\}|$ is the number of all labels ranked higher than label $j$, $w_j$ is the weight we put to label $j$ and $W_i$ is the sum of weights of all labels assigned to example $i$. For the MLC datasets, we put equal weights to all labels, whereas, for the HMLC datasets, we weighted each label with $0.75^d$, with $d$ being the depth of the label in the hierarchy (*Kocev et al., 2013*). For hierarchies that are directed acyclic graphs, the depth of a node is calculated as the average depth of its parent nodes plus one. The same weights are also used as the clustering weights for the targets for all methods.

## RESULTS AND DISCUSSION

### Predictive performance comparison

We first present the results obtained on the *rf2* dataset in Fig. 2. Here, the semi-supervised approach outperforms supervised learning for both SPYCT variants. This is the case in both single-tree and ensemble settings and for all considered numbers of labeled examples. These results demonstrate the potential of the proposed SSL methods.

For a high-level comparison of the predictive performance of the proposed SSL methods and the baselines, we use average ranking diagrams (*Demsar, 2006*). The results are presented in Fig. 3. The first observation is that SSL-SPYCT-GRAD achieves the best rank for all numbers of labeled examples in both single tree and ensemble settings. The only exception are single trees with 25 labeled examples, where it has the second-best rank, just slightly behind SSL-SPYCT-SVM. Additionally, SSL-SPYCT-SVM also ranks better than both its supervised variant and SSL-PCT for all values of $L$ and both single tree and ensemble settings. For standard PCTs, the semi-supervised version performed better than the supervised version in a single tree setting with very few labeled examples ($L = 25, 50$), otherwise, their performances were similar. This is consistent with the previous studies (*Levatić et al., 2017*; *Levatić et al., 2018*; *Levati et al., 2020*).

Next, we dig deeper into the comparison of SSL-SPYCT variants to the supervised SPYCTs and SSL-PCTs. We performed pairwise comparisons among the competing pairs with sign tests (*Demsar, 2006*) on the number of wins. An algorithm "wins" on a dataset if its performance, averaged over the 10 cross-validation folds, is better than the performance of its competitor. The maximum number of wins is therefore 30 (28 for $L = 500$). Tables 2 and 3 present the results for single tree and ensemble settings, respectively.

The results show that in the single tree setting, SSL-SPYCTs tend to perform better than their supervised counterparts, though the difference is rarely statistically significant. When used in ensembles, the improvement of the SSL-SPYCT-SVM variant over its supervised counterpart is small. With the gradient variant, the improvement is greater, except for the largest number of labeled examples. Compared to SSL-PCTs, the improvements are generally greater. This holds for both single trees and especially ensembles, where the differences are almost always statistically significant. As the average ranking diagrams in Fig. 3 already suggested, the gradient variant is especially successful.

Overall, the results also show that SPYCTs are a more difficult baseline to beat than SSL-PCTs. This is especially true in ensembles, where the studies of SSL-PCTs show that the improvement over supervised PCT ensembles is negligible (*Levatić et al., 2017*; *Levatić et al., 2018*; *Levati et al., 2020*). On the other hand, our results show SSL-SPYCT-GRAD can improve even the ensemble performance. Another important observation is that supervised variants never significantly outperform the SSL variants. This confirms that dynamically selecting the $\omega$ parameter prevents scenarios where unlabeled examples are detrimental to the predictive performance and supervised learning works better.

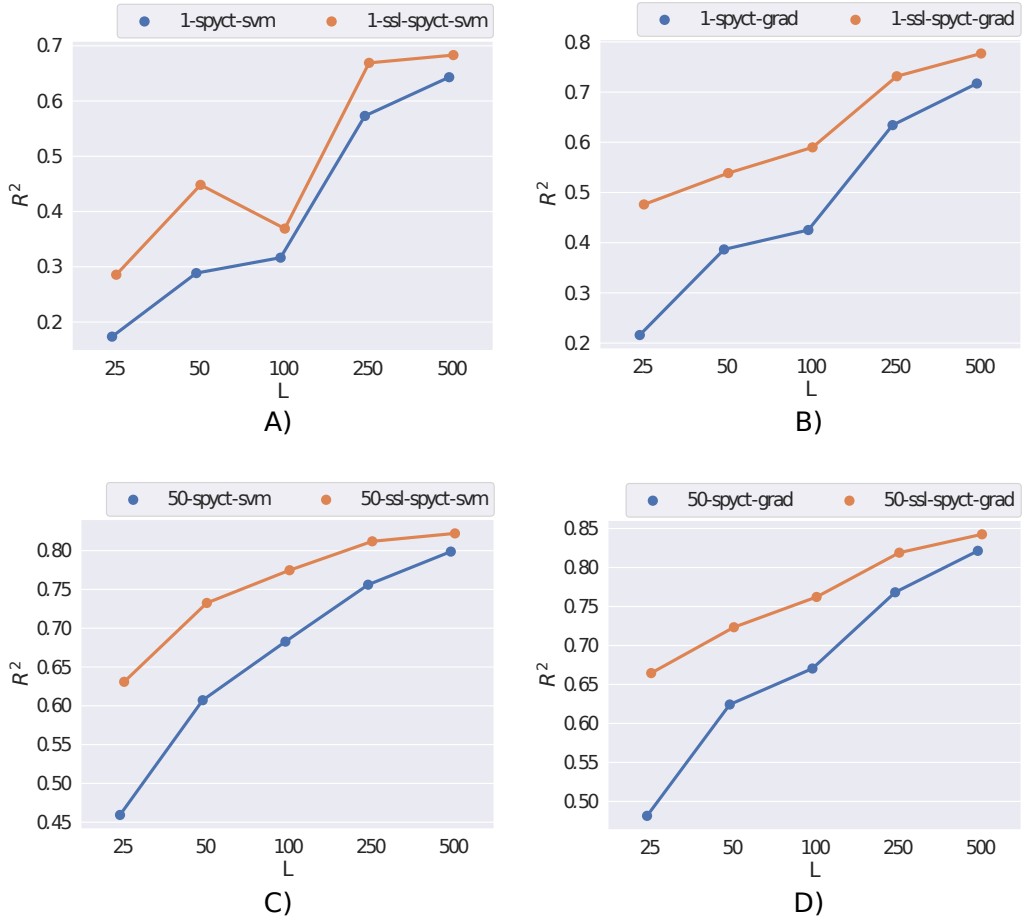

**Figure 2** Comparison of supervised and semi-supervised variants of SPYCT-SVM and SPYCT-GRAD methods (A & C, B & D) in both single tree and ensemble settings (A-B, C-D) on the *rf2* dataset with different numbers of labeled examples (*L*).

## Learning time comparison

To compare the learning times of the proposed SSL methods and SSL-PCTs, we selected one large dataset for each predictive task. We focused on the large datasets where the differences highlight the scalability of the methods with respect to the numbers of features and targets. We compare learning times of tree ensembles, as they also serve as a (more reliable) comparison for learning times of single trees. Fig. 4 shows the learning times on the selected datasets. The results confirm our theoretical analysis and show that the proposed SSL-SPYCTs are learned significantly faster than SSL-PCTs. The differences are especially large on datasets with many features and/or targets (e.g., *ara_interpro_GO*). The learning times are most similar on the *gesture* dataset, which has only 32 features, so the theoretical advantage of SSL-SPYCTs is less accentuated. Notwithstanding, the proposed methods are faster also on this dataset.

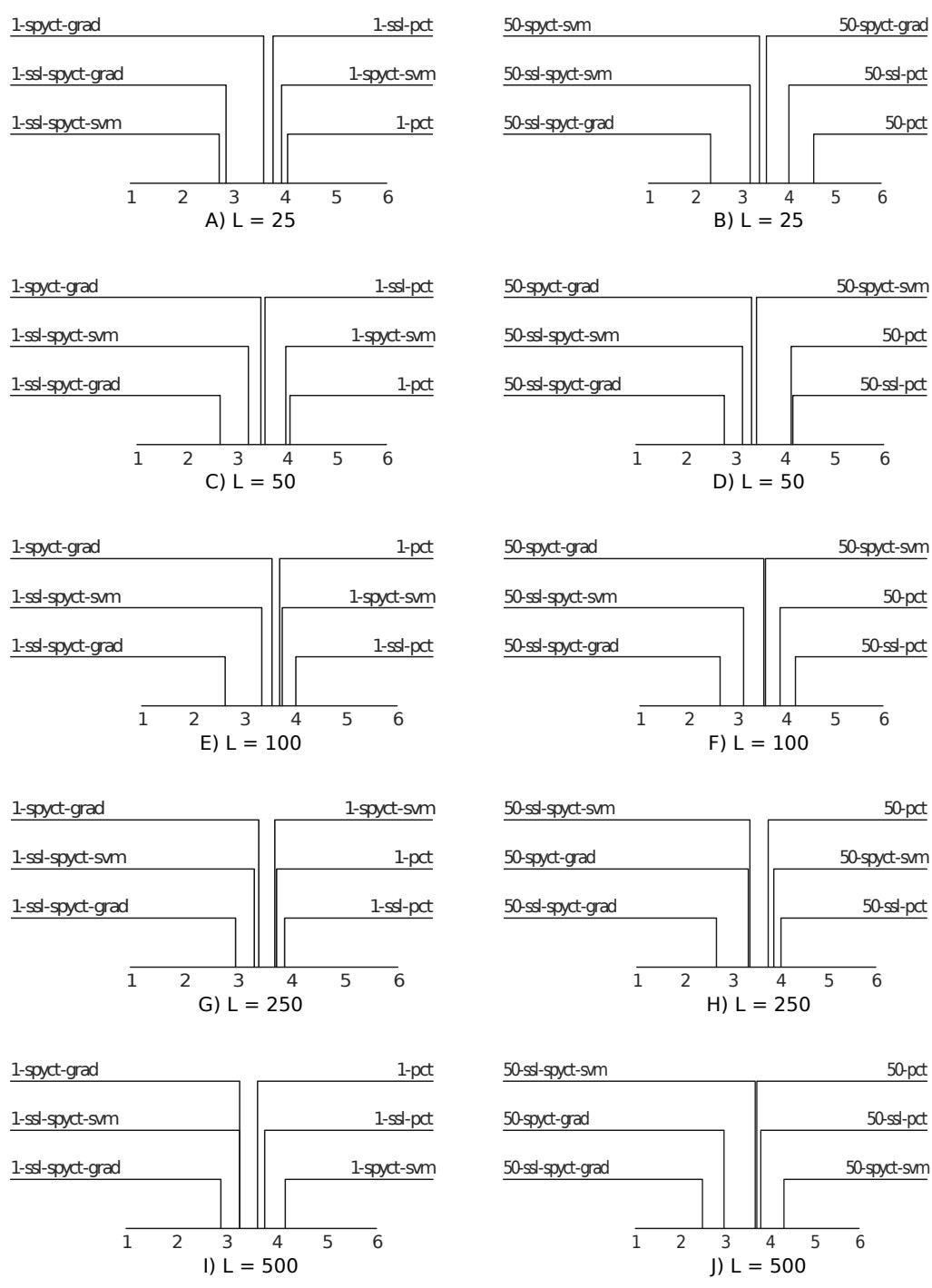

**Figure 3** Average ranking diagrams comparing the predictive performance of the proposed SSL-SPYCT-SVM (A, C, E, G, I) and SSL-SPYCT-GRAD (B, D, F, H, J) methods and the baselines with different numbers of labeled examples.

**Table 2  Comparison of the proposed SSL-SPYCT methods to their supervised counterparts and SSL-PCTs in single tree setting in terms of number of wins.** Bolded values indicate that the sign test Demsar06 showed significant difference in performance at $\alpha = 0.05$.

| #wins | $L = 25$ | $L = 50$ | $L = 100$ | $L = 250$ | $L = 500$ |
|---|---|---|---|---|---|
| 1-SSL-SPYCT-GRAD vs. 1-SPYCT-GRAD | 20 | 19 | **21** | 19 | 14 |
| 1-SSL-SPYCT-SVM vs. 1-SPYCT-SVM | 20 | 18 | 18 | 17 | 20 |
| 1-SSL-SPYCT-GRAD vs. 1-PCT-SSL | **21** | 18 | **22** | 18 | **20** |
| 1-SSL-SPYCT-SVM vs. 1-PCT-SSL | **22** | 18 | 18 | 19 | 16 |

**Table 3  Comparison of the proposed SSL-SPYCT methods to their supervised counterparts and SSL-PCTs in ensemble setting in terms of number of wins.** Bolded values indicate that the sign test Demsar06 showed significant difference in performance at $\alpha = 0.05$.

| #wins | $L = 25$ | $L = 50$ | $L = 100$ | $L = 250$ | $L = 500$ |
|---|---|---|---|---|---|
| 50-SSL-SPYCT-GRAD vs. 50-SPYCT-GRAD | 24 | 19 | 19 | 20 | 14 |
| 50-SSL-SPYCT-SVM vs. 50-SPYCT-SVM | 16 | 16 | 19 | 15 | 16 |
| 50-SSL-SPYCT-GRAD vs. 50-SSL-PCT | **25** | 22 | **22** | 21 | **21** |
| 50-SSL-SPYCT-SVM vs. 50-SSL-PCT | **23** | 21 | 22 | 21 | 15 |

## Investigating the $\omega$ parameter

The $\omega$ parameter controls the amount of influence of the unlabeled examples on the learning process. Fig. 5 shows the distributions of the $\omega$ values selected with the internal 3-fold cross-validation. We can see that the selected values varied greatly, sometimes different values were chosen even for different folds of the same dataset. This confirms the need to determine $\omega$ with internal cross-validation for each dataset separately. Additionally, we notice that larger $\omega$ values tend to be selected with more labeled examples and by ensembles compared to single trees. With larger numbers of labeled examples, it makes sense that the model can rely more heavily on the labeled part of the data and unlabeled examples are not as beneficial. For ensembles, this indicates that they can extract more useful information from few labeled examples compared to single trees. Whereas this seems clear for larger datasets, bootstrapping on few examples is not obviously beneficial. The fact that ensembles tend to select larger $\omega$ values (especially the SVM variant) also explains why the differences in predictive performance between supervised and semi-supervised variants are smaller in ensembles compared to single trees. We also investigated whether the selected $\omega$ values were influenced by the predictive modeling task (regression vs. classification, single target vs. multiple targets), but we found no noticeable differences between the $\omega$ distributions.

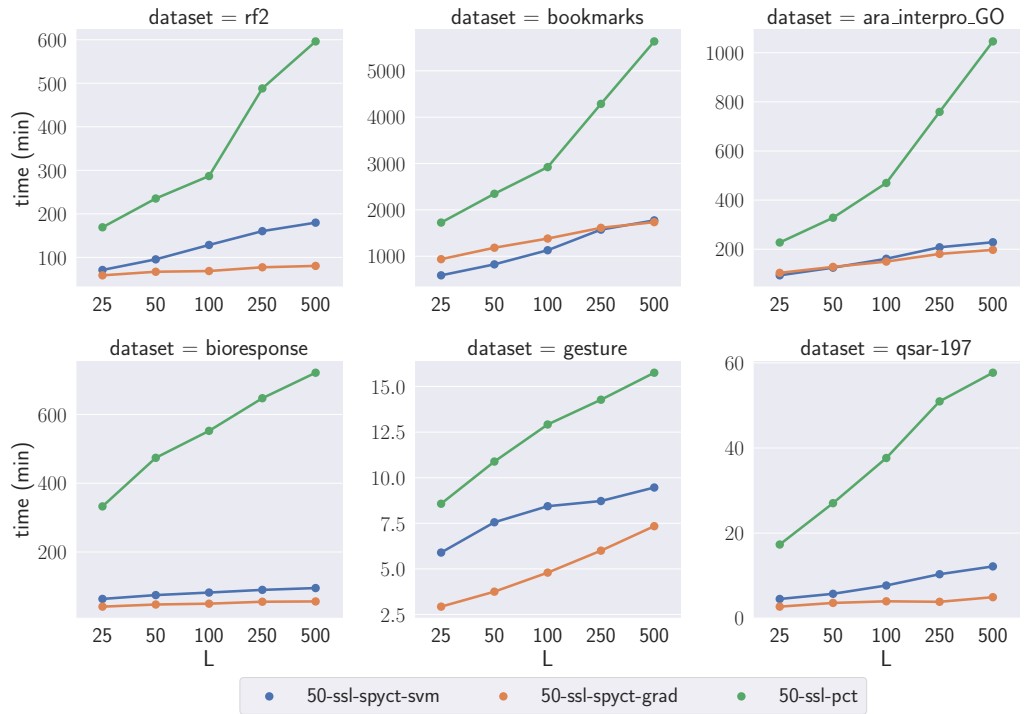

**Figure 4** Comparison of learning times of the SSL algorithms on a selection of large benchmark datasets.

## Investigating feature importances

We can extract feature importance scores from learned SPYCT trees (*Stepinik & Kocev, 2020*). The importances are calculated based on absolute values of weights assigned to individual features in all the split nodes in a tree (or ensemble of trees). For a single oblique PCT, they are calculated as follows:

$$imp(T) = \sum_{s \in T} \frac{s_n}{N} \frac{s_w}{\|s_w\|_1},$$

where $s$ iterates over split nodes in tree $T$, $s_w$ is the weight vector defining the split hyperplane, $s_n$ is the number of learning examples that were present in the node and $N$ is the total number of learning examples. The contributions of each node to the final feature importance scores are weighted according to the number of examples that were used to learn the split. This puts more emphasis on weights higher in the tree, which affect more examples. To get feature importance scores of an ensemble, we simply average feature importances of individual trees in the ensemble.

These scores tell us how much the model relies on individual features and can also be used to identify important features for a given task. We investigated if SSL-SPYCTs are more successful at identifying important features compared to supervised SPYCTs in problems with limited labeled data. To do this, we followed the setup from *Stepinik & Kocev (2020c)* and added random features (noise) to the datasets. For each original feature, we added a random one so that the total number of features was doubled. The values of the

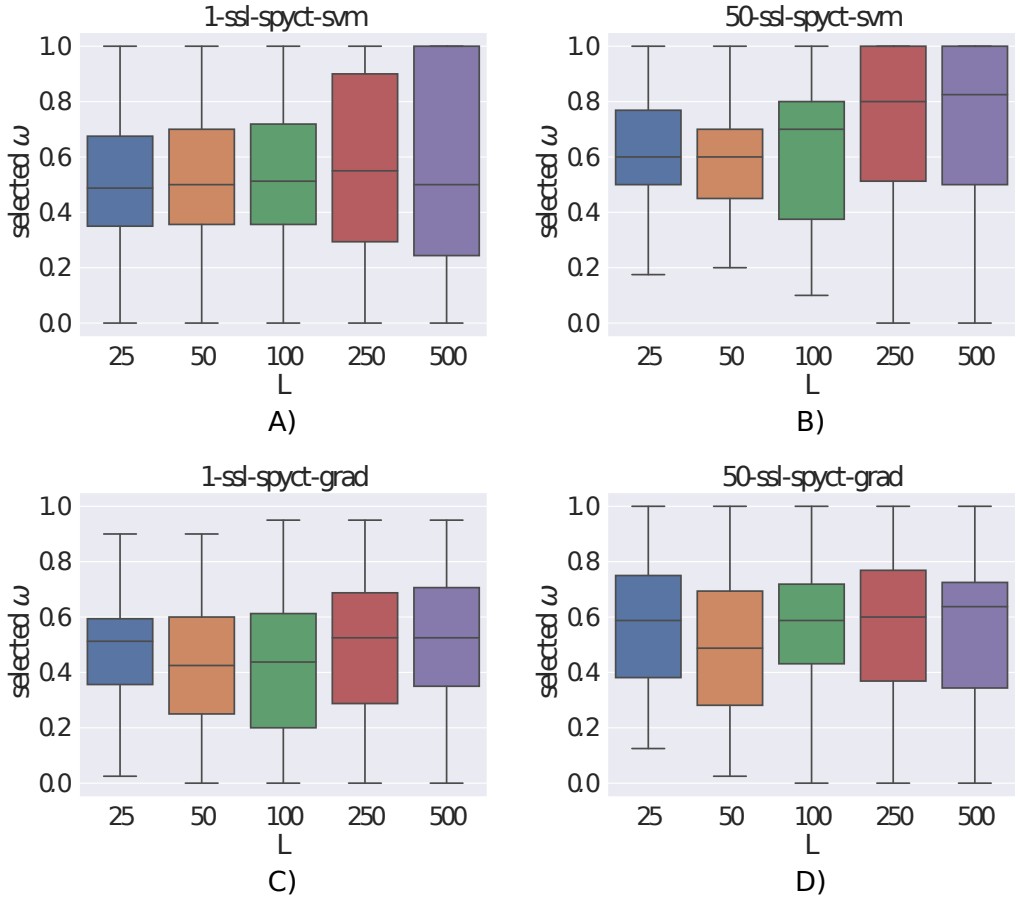

**Figure 5** (A-D) Distributions of the selected $\omega$ parameters for different number of labeled examples.

added features were independently sampled from a standard normal distribution. Then we learned SPYCTs and SSL-SPYCTs and compared the extracted feature importances.

Figure 6 presents the results on the qsar-197 dataset. For convenience, we also show the predictive performances of SPYCT and SSL-SPYCT methods. A good feature importance scoring would put the scores of random features (orange) to zero, whereas some real features (blue) would have noticeably higher scores. Low scores of many real features are not concerning, as datasets often include features that are not very useful for predicting the target. This example shows that SSL-SPYCTs can be better at identifying useful features than supervised SPYCTs. The difference here is greater with the gradient variant, especially with 50-250 labeled examples. This is also reflected in the predictive performance of the methods.

In general, the quality of feature importance scores obtained from a model was correlated with the model's predictive performance. This is expected and means that the conclusions here are similar. In terms of feature importance scores, SSL-SPYCTs are often similar to supervised SPYCTs, but there are several examples (e.g., Fig. 6) where they are significantly better and worth the extra effort.

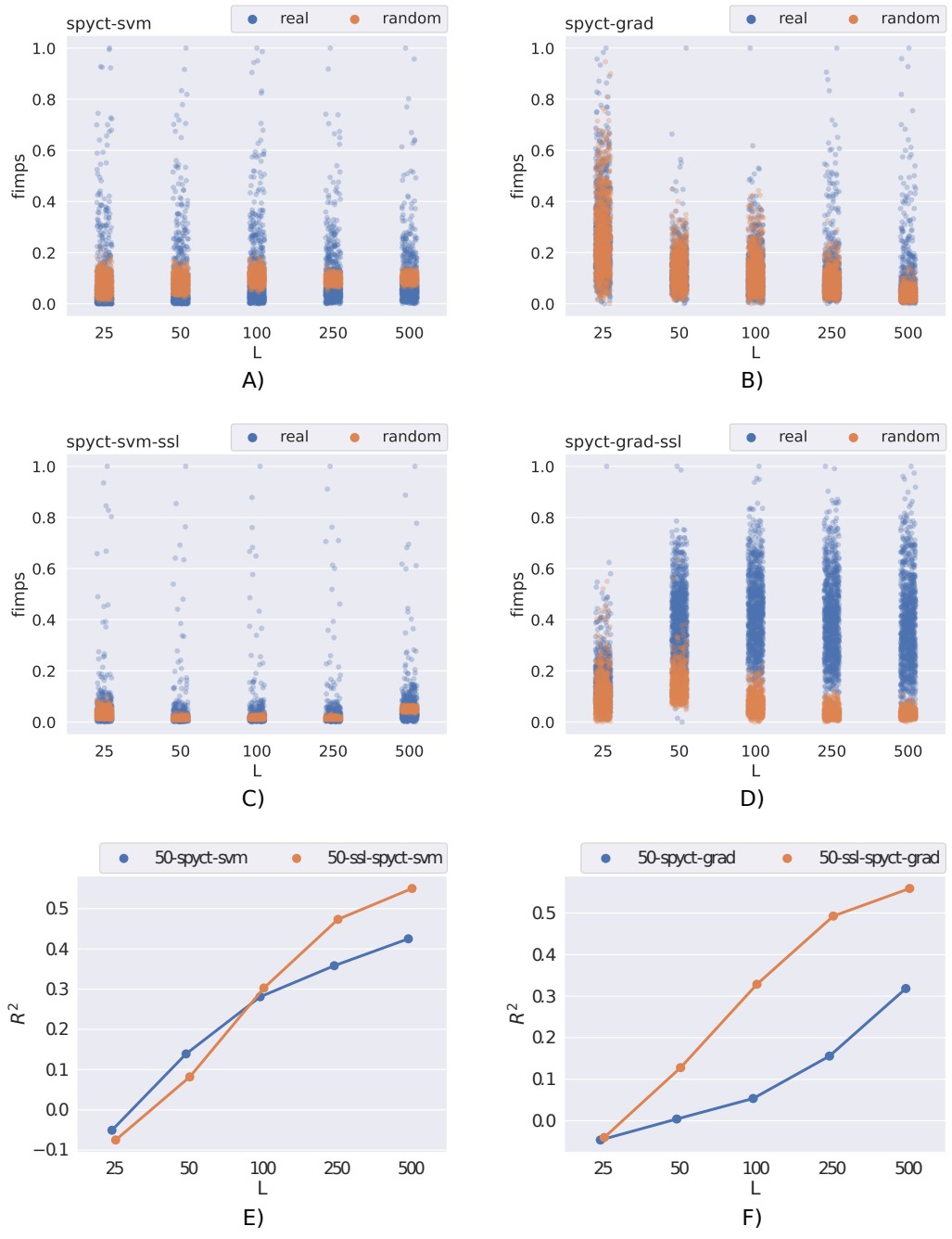

**Figure 6  Comparison of feature importance scores of real and random features (scaled to [0, 1] interval) for the qsar-197 dataset with different numbers of labeled examples.** (A, C, E) The SPYCT-SVM method, (B, D, F) the SPYCT-GRAD method. (A, B) Importance scores obtained with the supervised method, (C, D) the importance scores obtained with the unsupervised method, and (E, F) the predictive performance of both supervised and semi-supervised methods.

## CONCLUSION

In this paper, we propose semi-supervised learning of oblique predictive clustering trees. We follow the approach of standard semi-supervised predictive clustering trees and adapt both SVM and gradient variants of SPYCTs and make them capable of learning from unlabeled examples. The main motivation for the proposed methods was the improved computational scaling of SPYCTs compared to PCTs which is highlighted in the proposed SSL approach, where features are also taken into account when evaluating the splits.

We experimentally evaluated the proposed methods on 30 benchmark datasets for various predictive modeling tasks in both single tree and ensemble settings. The experiments confirmed the substantial theoretical computational advantage the proposed SSL-SPYCT methods have over standard SSL-PCTs. The results also showed that the proposed methods often achieve better predictive performance than both supervised SPYCTs and SSL-PCTs. The performance edge was preserved even in ensemble settings, where SSL-PCTs typically did not outperform supervised PCTs. Finally, we demonstrated that SSL-SPYCTs can be significantly better at obtaining meaningful feature importance scores.

The main drawback of SSL-SPYCTs (which is shared with SSL-PCTs) is the requirement to determine the $\omega$ parameter dynamically with internal cross-validation. This increases the learning time compared to supervised learning but prevents occasions where introducing unlabeled examples into the learning process hurts the predictive performance. We investigated the selected values for $\omega$ and found that higher values tend to be selected when there is more labeled data available, and by ensembles compared to single trees. But the selected values were still very varied, which confirms the need for dynamic selection of $\omega$.

For future work, we plan to investigate SPYCTs in boosting ensembles for both supervised and semi-supervised learning. Variants of gradient boosting (*Friedman, 2001*) have proven especially successful in many applications recently. We will also try improving the interpretability of the learned models with Shapley additive explanations (SHAP, *Lundberg et al. (2020)*). Because our method is tree-based we might be able to calculate the Shapley values efficiently, similarly to how they are calculated for axis-parallel tree methods.

### Funding

The work was supported by the Slovenian Research Agency through a young researcher grant and the grant J2-9230. The funders had no role in study design, data collection and analysis, decision to publish, or preparation of the manuscript.

### Grant Disclosures

The following grant information was disclosed by the authors:
The Slovenian Research Agency: J2-9230.

### Competing Interests

The authors declare there are no competing interests.

## Author Contributions

- Tomaž Stepišnik conceived and designed the experiments, performed the experiments, analyzed the data, performed the computation work, prepared figures and/or tables, authored or reviewed drafts of the paper, and approved the final draft.
- Dragi Kocev conceived and designed the experiments, analyzed the data, authored or reviewed drafts of the paper, and approved the final draft.

## Data Availability

Datasets from openml are available at the following links:

bioresponse - https://www.openml.org/d/4134

mushroom - https://www.openml.org/d/24

phoneme - https://www.openml.org/d/1489

spambase - https://www.openml.org/d/44

speeddating - https://www.openml.org/d/40536

cardiotocography - https://www.openml.org/d/1466

gesture - https://www.openml.org/d/4538

isolet - https://www.openml.org/d/300

mfeat-pixel - https://www.openml.org/d/40979

plants-texture - https://www.openml.org/d/1493

cpmp-2015 - https://www.openml.org/d/41700

pol - https://www.openml.org/d/201

qsar-197 - https://www.openml.org/d/3153

qsar-12261 - https://www.openml.org/d/3262

satellite_image - https://www.openml.org/d/294

MLC datasets are available from Mulan (below the table with their properties):

http://mulan.sourceforge.net/datasets-mlc.html. They can be identified by their names (bibtex, birds, bookmarks, delicious, scene).

MTR datasets from Mulan are all available in a single archive here:

http://mulan.sourceforge.net/datasets-mtr.html. Individual datasets (atp1d, enb, oes97, rf2, scm1d) can be identified by their names in the archive.

HMLC datasets are available at:

- https://dtai.cs.kuleuven.be/clus/hmc-ens/ one can find the datasets yeast_seq_FUN (labeled as D1) and ara_interpro_GO (labeled as D17).

- http://kt.ijs.si/DragiKocev/PhD/resources/doku.php?id=hmc_classification one can find the datasets imclef07d, diatoms, and enron. They can be identified by their names.

The implementation of the proposed method is also freely available at Gitlab:

https://gitlab.com/TStepi/spyct.

## Supplemental Information

Supplemental information for this article can be found online at http://dx.doi.org/10.7717/peerj-cs.506#supplemental-information.

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
