# Peer review of "Semi-supervised oblique predictive clustering trees"

_PeerJ Computer Science, doi:10.7717/peerj-cs.506_

## Round 0.1 · original submission · Major Revisions

Be especially careful with the experimental setup of the paper.

·

Basic reporting

- The proposal is in overall good as well as the organization of the paper. I greatly appreciate the fact of making the code of the tool publicly available through git repository.
- Please, provide vectorial images (eps, pdf, ...) when possible. Furthermore, for most of the figures (such as figures 4 and 5), authors should include a legend for lines/colors to improve their readability.
- Please, be consistent with the notation. In table 2 methods are called "SPYCT-GRAD-SSL", while in Table 3 they are called "SSL-SPYCT-GRAD"

Experimental design

- Authors repeat the experiments 10 times with each configuration. It would be desirable to repeat at least 30 times for each different configuration, in order to obtain more consistent results in statistical tests.
- In Figure 3 as well as in Tables 2-3, authors include the results for all algorithms. These results are for all datasets, each of them averaged among their 10 (hopefully 30 in the future) executions? Please, make it clearer.
- I encourage authors to make raw results publicly available to ease the reproducibility of results in any kind of supplementary material (as a website or in the gitlab repository). At least those averaged results among all executions, not necessary the results of each single execution.
- When studying the learning time, authors only report the results on big datasets. I think it would be a good approach to illustrate the results in a figure (as they do), but results for all datasets should be reported and studied. If the proposed method do not perform faster in smaller datasets but does in bigger, it should be stated (or if it run faster in all/most cases). Besides, I would encourage to include the times of single trees too.

Validity of the findings

- The validity of the results is great; the proposed method perform significantly better than state-of-the-art in some cases, but never significantly worse. Besides, their proposed method run much faster than state-of-the-art.

Additional comments

- The paper is in overall good, but some improvements should be made in the experimental setting to finally accept the paper.

Reviewer 2 ·

Basic reporting

In general, the paper is well written and related literature is well cited. Figures and tables are adequate.

For reporting the results, the manuscript only contains ranking diagrams and the number of wins per method. To allow a better understanding of how good your method is, I would like to suggest the inclusion of the numerical values themselves (perhaps in an appendix or supplemental material).

Small remarks: figure 4 is missing a key and the symbol 'w' is used with different meanings (e.g. rule 121 versus rule 129).

Experimental design

The evaluation metrics for multi-target regression (MTR) and hierarchical multi-label classification (HMLC) are somewhat unconventional. What is the advantage of using the coefficient of determination for MTR as opposed to the more common average relative root mean squared error? For HMLC, why exactly was the LRAP measure chosen, has it been used before?

The current evaluation for feature importance is rather unclear. It relies on adding noisy features and comparing their scores against real features, nonetheless this raises several comments.
a) What exactly do you mean by score?
b) How different coefficients value affect your score? I assume that the same feature may have very different coefficients depending on the split, can they cancel each other out?
c) How are these random features generated?

As reported in Kocev 2013, targets with different ranges can affect the heuristic values differently. This is solved by normalizing the targets using their variance beforehand. Does your implementation take that into account?

Validity of the findings

My main concern regarding the validity of the results is the transductive experimental set-up for the proposed approach: if the unlabeled training set is used as test set, it seems to me somewhat an unfair comparison to the supervised setting. In my opinion, either the predictive performance should be compared on an independent test set for both SSL and supervised settings (i.e., inductive) or the learning task should be clearly defined in the method section (e.g., when introducing the notation in the first paragraph).

In the ensemble setting, bagging is used. Indeed, with bagging you will have a time complexity quadratic in the number of features. However, more common ensembles like random forests will also reduce time complexity. How does bagging with oblique splits correspond to random forests with default splits in learning time (and predictive performance)?

For reproducibility purposes, parameter values should be included. Please find a below a list of parameters that I have failed to find in the manuscript:
- C -> SVM and gradient variant;
- initial w, initial b and Adam parameters -> gradient variant;
- Stopping criteria to grow trees -> I could only find the following quote
"The splitting of the examples (i.e., the tree construction) stops when at least one of the following stopping criteria is reached. We can specify the minimum number of examples required in leaf nodes (at least one labeled example is always required otherwise predictions cannot be made). We can also require a split to reduce the impurity by a specified amount or specify the maximum depth of the tree"
- Label weights for the ara_interpro_GO dataset: -> I could only find the excerpt:
"we weighted each label with 0.75^d, where d is the depth of the label",
nonetheless a label can have multiple values for d since it may have several parent nodes. How do you define the weights here?

I appreciate the effort of providing the source code to your method, nonetheless, to make it even further reproducible, I would suggest including the data partitions as well.

Additional comments

The novelty of this work is rather limited. The proposed method is a direct combination of two already proposed methods in the literature by the same research group, namely oblique predictive clustering trees and semi-supervised predictive clustering trees. However, as I understand it, novelty should not play a role in the evaluation for this journal.

What is the size of trees generated by your method? Since your method allows multiple features per split, I assume that fewer splits should be enough to build a model, nonetheless that should come with a trade-off in interpretability and performance. For instance, how does one oblique split with 10 features in its hyperplane compare with a decision tree with 10 nodes on its path? I recommend the authors to provide a comparison between the number of features on average involved in performing a prediction.

Finally, I have some concerns about the motivation for the paper.
What are exactly the advantages of employing a linear of combination of features rather than a single one? In the manuscript, I found the excerpt: "The potential advantage of oblique splits compared to axis-parallel splits is presented in Figure 1. What do the authors mean exactly? I can notice that linear combinations may result in smoother splits, but how is that superior? Would you mind elaborating more on that?

---

## Round 0.2 · accepted · Accept

This new version meets all the requirements to be accepted. Congratulations.

·

Basic reporting

The authors have addressed all my comments in previous review.

Experimental design

The authors have addressed all my comments in previous review. The experimental design is now correct.

Validity of the findings

The validity of the results is great; the proposed method performs significantly better than the state-of-the-art in some cases, but never significantly worse. Besides, their proposed method run much faster than state-of-the-art.

Additional comments

The paper is overall good. I recommend accepting the paper as is.